# Asymmetric dynamics of DNA entering and exiting a strongly confining nanopore

Nicholas A.W. Bell[1], Kaikai Chen [1,2], Sandip Ghosal[3], Maria Ricci[1] & Ulrich F. Keyser [1]

In nanopore sensing, changes in ionic current are used to analyse single molecules in solution. The translocation dynamics of polyelectrolytes is of particular interest given potential applications such as DNA sequencing. In this paper, we determine how the dynamics of voltage driven DNA translocation can be affected by the nanopore geometry and hence the available configurational space for the DNA. Using the inherent geometrical asymmetry of a conically shaped nanopore, we examine how DNA dynamics depends on the directionality of transport. The total translocation time of DNA when exiting the extended conical confinement is significantly larger compared to the configuration where the DNA enters the pore from the open reservoir. By using specially designed DNA molecules with positional markers, we demonstrate that the translocation velocity progressively increases as the DNA exits from confinement. We show that a hydrodynamic model can account for these observations.

[1] Department of Physics, Cavendish Laboratory, Cambridge University, CB3 0HE Cambridge, UK. [2] Department of Mechanical Engineering, State Key Laboratory of Tribology, Tsinghua University, Beijing 100084, China. [3] Department of Mechanical Engineering and Engineering Sciences and Applied Mathematics, Northwestern University, Evanston, IL 60208, USA. Nicholas A. W. Bell and Kaikai Chen contributed equally to this work. Correspondence and requests for materials should be addressed to U.F.K. (email: ufk20@cam.ac.uk)

The transport of polymers through nanopores is an area of great scientific interest that combines ideas from electro-kinetics, polymer dynamics and fluid mechanics[1–9]. The translocation of DNA provides a model system for studying the question of how a uniformly charged polymer threads through a nanopore. There is also significant technological impetus for understanding DNA transport given potential applications such as DNA mapping and sequencing[10–13]. Despite recent progress, quantitative experiments on the dynamics of the translocation process, in particular the trajectory profiles of DNA passing through solid-state nanopores, are still lacking[14]. There is also a deficit in our understanding of how different nanopore geometries can affect translocation dynamics and which geometry gives the highest resolution possible for reading information along the DNA contour[15, 16].

Solid-state nanopores can be fabricated in a wide variety of three-dimensional (3D) geometries and therefore provide an ideal platform for investigating the effects of nanopore geometry on translocation. Early experiments on solid-state nanopores used two-dimensional (2D) free standing membranes 20 nm or less in thickness[17–19]. Pores can be fabricated in these membranes by ion beam ablation or voltage breakdown of materials such as silicon nitride or graphene[20, 21]. Alternatively, long conically shaped nanopores with controllable taper angles can be fabricated using laser-assisted capillary pullers or by etching of heavy ion tracks[22, 23]. The translocation properties of DNA into a conically shaped nanopore show many similarities with translocations for the 2D membrane case when one takes into account the longer effective sensing length[24, 25]. Recently, several strategies have been investigated for modifying geometric constraints on translocating polymers by integrating porous fibre networks or agarose gels on one side of a membrane[26, 27]. This is in part motivated by the need to reduce the velocity of DNA translocation thereby potentially improving resolution. The geometry of the pore is known to also play a role in the transport dynamics of single stranded DNA through biological pores. For instance, the DNA capture rate and current signature during passage through α-hemolysin depends on the direction of transport due to the pore's structural asymmetry[28, 29].

Here, we demonstrate how DNA translocation speed can be affected by solid-state nanopore geometries which create restrictions on the available DNA conformations. We employ asymmetric conical nanopores and investigate the translocation dynamics into and out of confinement. The slow tapering angle provides a strong quasi-one-dimensional (1D) confinement thereby limiting the available DNA coil conformations. We find that the average translocation time is significantly greater when exiting the conical pore than when entering from the reservoir. Furthermore, by using a custom designed DNA ruler with multiple position markers, we show that the DNA speeds up significantly during the time course of translocation out of confinement. Using an experimental set-up combining a nanopore with optical tweezers, we measure the tether force required to stall the DNA translocation. We present a physical model of translocation based on nanopore hydrodynamics and DNA elasticity that utilises the measured tether force and provides a quantitative model for the translocation process. Our findings demonstrate a comprehensive understanding of DNA dynamics through conical nanopores.

## Results

**Translocation configurations in confinement.** Figure 1a shows a scanning electron microscope image of a conically shaped nanopore fabricated by laser-assisted capillary pulling. Such conical pores have previously been used to study the transport

of ions[30, 31] and macromolecules[32–34] in confinement. The diameter of the nanopores used here was estimated as $14 \pm 3$ nm (mean ± s.d.) with a cone semi-angle of $0.05 \pm 0.01$ radians (mean ± s.d.) based on previous characterisation of the fabrication process[16]. All experiments were performed in 4 M LiCl electrolyte. The direction of translocation is determined by the polarity of the voltage applied as shown schematically in Fig. 1b, c. By switching the voltage immediately after the DNA crosses the pore (indicated by the return of the current to the baseline value), we are able to measure translocation speeds in the two opposite directions (Fig. 1c, d); a so-called ping-pong experiment[35]. In this paper, we will use the phrase forward translocation to indicate the entry of DNA from the reservoir into confinement and backward translocation when the DNA travels in the opposite direction. Hundreds of translocations were recorded from these recapture experiments to enable statistical analysis.

Figure 2a shows a scatter plot of the current change during translocation against event duration in either direction using 8 kbp DNA. Backward translocation shows a notably wider scatter in the current signal and substantially longer event durations. We measured the same effect when translocations were

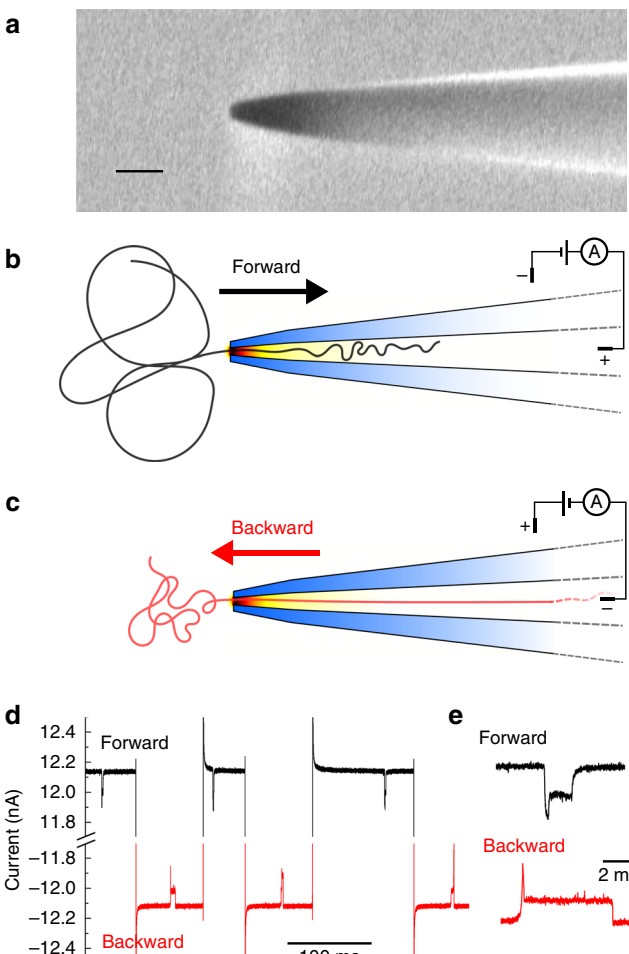

**Fig. 1** Set-up for DNA translocations and examples of the ping-pong experiment. **a** Example scanning electron microscope image showing the outer dimensions of the conical nanopore. *Scale bar = 100 nm.* **b** Forward translocation: DNA enters the nanopore from the open reservoir. **c** Backward translocation: DNA exits the nanopore and passes into the reservoir. **d** Ionic current trace showing multiple translocations of a single DNA molecule which is moved forward and backward through the pore by repeatedly switching the polarity of the 600 mV potential. **e** Current traces in forward and backward translocation at greater temporal resolution

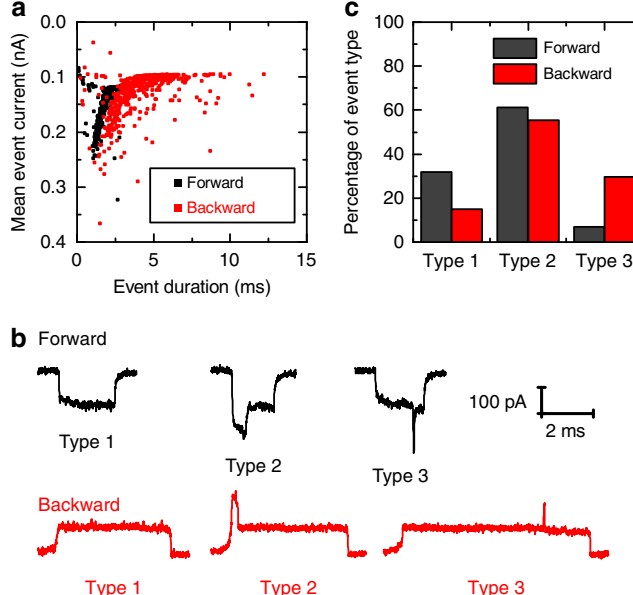

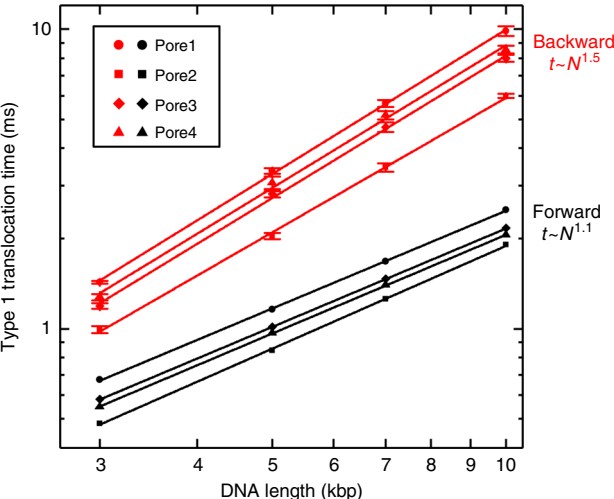

**Fig. 2** Comparison of scatter plot and translocation types observed for forward and backward directions. **a** Scatter plot showing statistics of forward and backward translocations for 8 kbp DNA recorded in ping-pong experiments using 570 events. **b** Typical examples of events categorised into three types: Type 1 showing only one level, Type 2 showing a fold at the event beginning and Type 3 showing no beginning fold but a later deviation. **c** Relative frequency of occurence of the three event types in forward and backward translocations

**Fig. 3** Type 1 translocation time as a function of DNA length. Data show mean value, *error bars* in the backward direction are the standard error of the mean (*error bars* are smaller than symbol sizes in forward direction). Each data set shows a least squares linear fit to the data. The average scaling exponent of the four pores is displayed

driven by a constant potential as when using the ping-pong method indicating that the fast recapturing does not affect the result significantly (Supplementary Fig. 1). This can be expected given that we switch the voltage 40 ms after detecting an event which is significantly more than the Zimm relaxation time of 8 kbp DNA in 4 M LiCl (~8 ms)[36]; thus, the DNA chain will have had sufficient time to equilibrate after translocation[37]. In contrast to the results shown here for conical nanopores, ping-pong experiments with nanopores in thin 2D membranes show the same translocation time irrespective of which direction the DNA is driven through as expected for two symmetric semi-infinite spaces[35]. In Fig. 2a, b we quantify the different types of translocation events according to the shape of the ionic current blockade signal. The shape of the blockade gives an indication of the configuration of the molecule as it passes through the nanopore. We classified the events into three categories; events which show only one level (type 1), events which show a fold at the beginning (type 2) and events which do not show a fold at the beginning but show a significant deviation from the type 1 level later in the translocation (type 3). The exact quantitative definitions of these classifications are given in Supplementary Fig. 2. Event types in the forward direction have been extensively characterised for conical quartz nanopores[24] and show very similar behaviour to that seen in solid-state nanopores in 2D membranes[38–40]. In agreement with these earlier results, we observe that for the 8 kbp DNA tested, the events are predominantly type 1 or type 2 with just 6% being type 3 signals. The configuration of the DNA in type 3 events was recently suggested to be mainly a result of knots in the DNA[41]. Interestingly, we observe (Fig. 2d) that there are a substantially higher fraction of these type 3 events in the backward direction. We hypothesise that these extra type 3 events are due to the DNA being transiently trapped in a multi-folded configuration which has to be resolved before translocation.

**Kinematics of translocation**. Having categorised the three event types observed, we further investigated the differences in translocation times for the forward and backward directions. We restricted our analysis to type 1, that is, completely unfolded translocations, by setting a suitable threshold for the ionic current level. The translocation times of a mixture containing four different DNA lengths (3, 5, 7 and 10 kbp) were measured in both translocation directions using ping-pong experiments. Using the ping-pong approach[35] enabled us to determine the DNA length from the event charge deficit in the forward direction[22]—the event scatter in the backward direction being too broad to be able to identify the different DNA lengths. In Fig. 3 we plot the mean translocation time as a function of DNA length in the two directions using data from a total of four nanopores. The translocation time, for a fixed direction and length, varies slightly from one pore to the next which we attribute to small differences in the geometry. We determined a power law for the average translocation time ($t$) with DNA length ($N$): $t \sim N^{\alpha}$ by least squares fitting. The average exponent is $\alpha = 1.1$ in the forward direction. For the backward translocations, we measure that the DNA translocation time is longer for every DNA length and the exponent is markedly higher with an average value of $\alpha = 1.5$.

In order to gain a better understanding of the kinematics of translocation, we used a recently developed method for tracking intra-event velocity with a DNA based ruler[36] (Fig. 4a). The DNA ruler contains six zones of protruding DNA dumbbell hairpins which are positioned at known intervals along the DNA contour. Translocations of this DNA ruler were recorded in ping-pong experiments. DNA fragments and translocations with a fold at the beginning (type 2) were filtered out (Supplementary Note 1) prior to further analysis of the data. We then used a peak finding algorithm to select only events which showed exactly six peaks corresponding to the six intra-chain markers. The event rejection rate for backward translocation was higher due to the propensity of type 3 events as shown in Fig. 2 (Supplementary Note 1). Examples of current signatures from the library of accepted events are depicted in Fig. 4b for backward, as well as forward translocations. The total DNA translocation time is larger in the backward direction as seen earlier (Fig. 3). It is also evident from Fig. 4b that in the backward direction the DNA tends to speed up significantly whereas in the forward direction the velocity remains

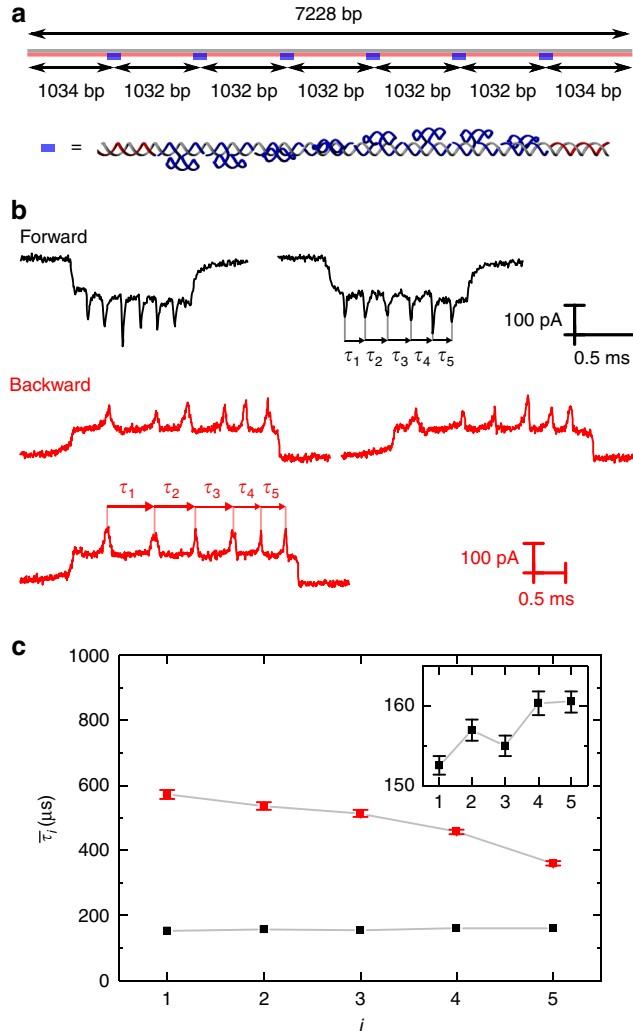

**Fig. 4** Intra-event velocity dynamics in forward and backward directions. **a** Design of construct with six zones containing DNA dumbbell hairpins separated by equal 1032 bp intervals. **b** Example translocations in the forward and backward directions. The times are measured between peaks as indicated. **c** Comparison of mean intra-event times in the two directions. Data are shown from a single nanopore ($N = 335$ forward and $N = 200$ backward). *Error bars* represent standard error of the mean. *Grey lines* connect the mean values and are used as a guide to the eye. *Inset* shows the forward direction data with a smaller *y*-axis range so that the error bars are visible

approximately constant albeit with a small increase of ~5% from $\tau_1$ to $\tau_5$ in agreement with previous results[36]. To quantify the intra-event dynamics, we measured the time between the markers and computed the average from several hundred events. Figure 4b shows the definitions of the intra-event intervals $\tau_i$ and in Fig. 4c the average values are plotted.

**Force measurements in stalled translocation.** An important quantity for understanding translocation dynamics is the force acting on the DNA during translocation[42]. We directly measured the force for our nanopore geometry and salt conditions by using an experiment combining optical tweezers and ionic current measurement. One end of the DNA ruler shown in Fig. 4a was modified with a biotin labelled oligonucleotide and subsequently conjugated to streptavidin coated 2 µm diameter spherical polystyrene beads. An individual bead was trapped with an optical tweezer while the ionic current and displacement from the trap

centre were simultaneously measured (Fig. 5a). The bead was brought towards the nanopore by a piezoelectric control stage and after a short time a reduction in current, and, concurrently, a displacement of the bead from the trap centre was observed (Fig. 5b). The bead was then retracted from the nanopore tip at a velocity of 115 nm/s. During the retraction, the ionic current showed further decreases as the dumbbell hairpins passed through the nanopore. The force trace also shows small perturbations due to the increased force when the dumbbell hairpins are present within the nanopore. We calculated the mean force by fitting a Gaussian function to force traces recorded from multiple insertions and retractions. The force was measured to be linear in the applied voltage as observed previously[43] with a gradient of 5.6 fN/mV. The force is significantly lower than measurements with similar sized nanopores in 1 M KCl which measured ~50 fN/mV[44]. This is consistent with the known decrease of DNA velocity in 4 M LiCl by a factor of ~10 compared to 1 M KCl. Both effects may be ascribed to a significant reduction in DNA effective charge due to the smaller size of Li$^+$ cations compared to K$^{+}$[45].

**Hydrodynamic model of translocation process.** Our experimental characterisation provides a detailed overview of translocation viz. the length dependence of translocation time, intra-event velocities in the two directions and the force acting on the DNA. To explain our observations we have developed a model for the DNA translocation using a continuum formulation treating ion transport by the Nernst–Planck formalism and hydrodynamics using the equations of Stokes flow. Though our system is not far removed from molecular scales, the continuum approach has been shown to be applicable to the length scales considered here[34, 46–50].

The hydrodynamic force on the DNA may be regarded as a vector sum of the forces acting on a large number of individual sections, each considered as an infinitesimal cylinder moving through the fluid. Thus, only relatively straight portions of the DNA contribute to the net force, the contributions from the fluctuating parts being small due to many cancellations in the vector sum. The net hydrodynamic force along the translocation axis may therefore be calculated by replacing the DNA within the conical pore by a straight cylinder of effective length $L$, where $L$ could be smaller than the true contour length of the DNA strand. The electric component of the force however is directed along the electric field vector irrespective of the orientation of individual DNA segments. The intensity of the electric field reaches its maximum magnitude at the pore entrance, decaying rapidly in space (as the inverse square of distance), while its direction is parallel to the axis of symmetry of the pore. Therefore, the DNA may be regarded as an infinitely long cylinder placed along the pore axis when calculating the net electric force. The electric force contribution from outside the nanopore is negligible compared to the contribution from within the nanopore (Supplementary Note 2).

Thus, we adopt the model of DNA as a uniformly charged rigid rod translating with velocity $v$ along the axis of the conical pore. We use cylindrical co-ordinates with origin at the apex of the cone (Supplementary Note 2). The co-ordinate $x$ is used to denote distance from the origin along the central axis and $r$ is the radial distance from the axis. In the lubrication limit[51] with thin Debye layers, the Stokes equation for fluid flow $u(r, x)$ between the DNA and pore is given by

$$u(r,x) = -\frac{\epsilon \zeta_s E(x)}{\mu} + \left\{ v + \frac{\epsilon \left( \zeta_s - \zeta_p \right)}{\mu} E(x) \right\} \ln\left(\frac{r}{\alpha x}\right) / \ln\left(\frac{a}{\alpha x}\right),$$

$$(1)$$

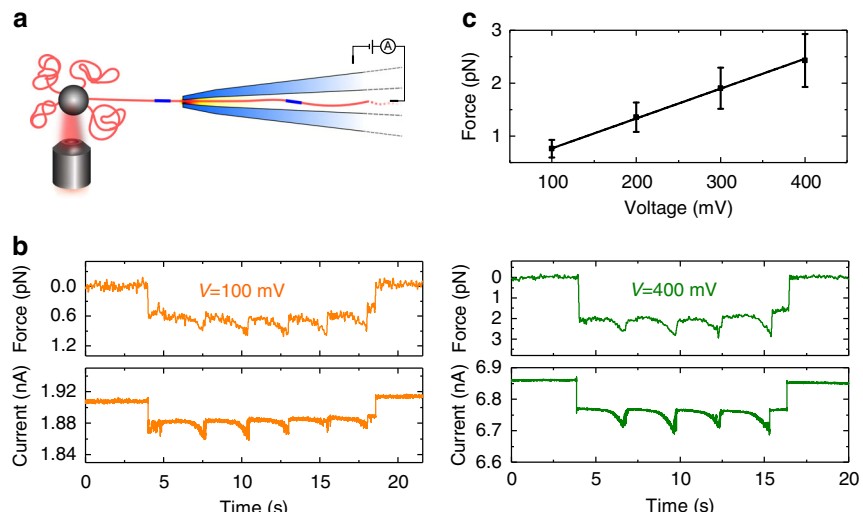

**Fig. 5** Determination of force on DNA using optical tweezers. **a** Schematic showing DNA stalled in a tug of war with the force from the optical trap equalling the combined force due to electrophoresis on the DNA and drag from electroosmosis. **b** Example traces showing simultaneous force and current recordings at 100 and 400 mV. After the DNA enters the nanopore, the colloid was immediately retracted at a speed of $v = 115$ nm/s. The number of peaks due to dumbbell sections is determined by how close the colloid is initially to the nanopore when the DNA is stalled. The force traces were filtered at 10 Hz and the current traces at 100 Hz for display. The force decreases slightly towards the end of the trace since the DNA no longer occupies the high electric field zone which extends a few hundred nanometres from the pore entrance[59]. **c** Force vs. voltage relationship. *Data points* show the mean value of a Gaussian fit to an all points histogram with the *error bar* showing the standard deviation of the fit. The gradient of the line is 5.6 fN/mV

### Table 1 Values of experimental parameters

| Experimental parameter | Value | Unit |
|---|---|---|
| Applied voltage ($V$) | 600 | mV |
| Pore radius ($R_0$) | 7 | nm |
| Cone semi-angle ($\alpha$) | 0.05 | Radian |
| Viscosity of electrolyte[60] ($\mu$) | 0.0017 | Pa s |
| DNA radius ($a$) | 1.0 | nm |
| DNA length ($L_{max}$) | 2,458 | nm |
| optical stall force ($F_{tether}$) | 3.4 | pN |

$R_0$ and $\alpha$ are average values determined by scanning electron microscopy. The DNA length is estimated based on 0.34 nm/bp helical rise. The tether force is extrapolated from the linear fit of Fig. 5 to the voltage (600 mV) used for translocations. The extrapolation also corrected for a small offset at 0 mV in Fig. 5

where $u(r, x)$ satisfies the Helmholtz-Smoluchowski slip boundary conditions at the nanopore wall and on the DNA surface:

$$u(r = \alpha x, x) = -\frac{\epsilon \zeta_s E(x)}{\mu} \qquad (2)$$

$$u(r = a, x) = v - \frac{\epsilon \zeta_p E(x)}{\mu}, \qquad (3)$$

here $\epsilon$ is the solution permittivity, $\zeta_s$ is the zeta potential of the substrate (quartz), $\zeta_p$ is the zeta potential of the charged polymer (DNA), $\mu$ is the viscosity of the electrolyte and $E(x)$ is the local electric field determined from the condition of current continuity. Definitions of symbols with values for the experiment are given in Table 1.

The translocation velocity is determined from the requirement that the net force on the DNA ($F$) must be zero (Supplementary Note 2):

$$F = -2\pi\mu \left[ v \int_{x_0}^{x_0+L} \frac{\mathrm{d}x}{\ln(\alpha x/a)} + v_{e0} \int_{x_0}^{\infty} \frac{x_0^2}{x^2 \ln(\alpha x/a)} \mathrm{d}x \right], \qquad (4)$$

where $v_{e0} = -\epsilon(\zeta_p - \zeta_s)\alpha V/(\mu R_0)$, $x_0 = R_0/\alpha$ and the upper limit in the second integral representing the electric contribution has been

set to infinity on account of the rapid decay of the electric field with distance. The first term of this equation represents the hydrodynamic drag on an uncharged cylinder moving with velocity $v$. The second term represents the resultant force from the electric driving force on the DNA and the viscous drag of the oppositely streaming adjacent counterion cloud, both of these effects are proportional to the applied voltage. Importantly the integrand of the first term due to viscous drag scales as $\sim 1/\ln(x)$ whereas the integrand of the second term due to the electric field scales as $\sim 1/x^2$ (neglecting a slowly varying logarithmic factor). This shows that the electric field is concentrated at the pore entrance whereas the hydrodynamic drag is effective over a much longer region. Equation (4) can be re-expressed as:

$$F = -2\pi\mu v \int_{x_0}^{x_0+L} \frac{\mathrm{d}x}{\ln(\alpha x/a)} + \lambda_e V = 0 \qquad (5)$$

where

$$\lambda_e = \frac{2\pi\epsilon\left(\zeta_p - \zeta_s\right)R_0}{a} \int_{R_0/a}^{\infty} \frac{\mathrm{d}\xi}{\xi^2 \ln \xi} \qquad (6)$$

may be regarded as an effective DNA charge which incorporates the electroosmotic drag from the counterions. This effective DNA charge may be determined from our force measurements using the optical tweezers. Since $\lambda_e$ is independent of $v$, in the presence of a tether force ($F_{tether}$) from the optical trap that immobilises the DNA, $-F_{tether} + \lambda_e V = 0$. Thus, the translocation velocity may be expressed solely in terms of measured quantities:

$$v = \frac{\alpha F_{tether}}{2\pi\mu a} \left[ \mathrm{li}\left(\frac{\alpha L}{a} + \frac{R_0}{a}\right) - \mathrm{li}\left(\frac{R_0}{a}\right) \right]^{-1}, \qquad (7)$$

using the logarithmic integral $\mathrm{li}(x) = \int_0^x \frac{\mathrm{d}\xi}{\ln \xi}$.

In order to make use of Eq. (7), we need to make some assumptions about the DNA effective length $L$, which is not known precisely. Outside the nanopore the DNA has no geometric constrictions and forms a random coil. Therefore, for short DNA molecules such as those used here, only a section

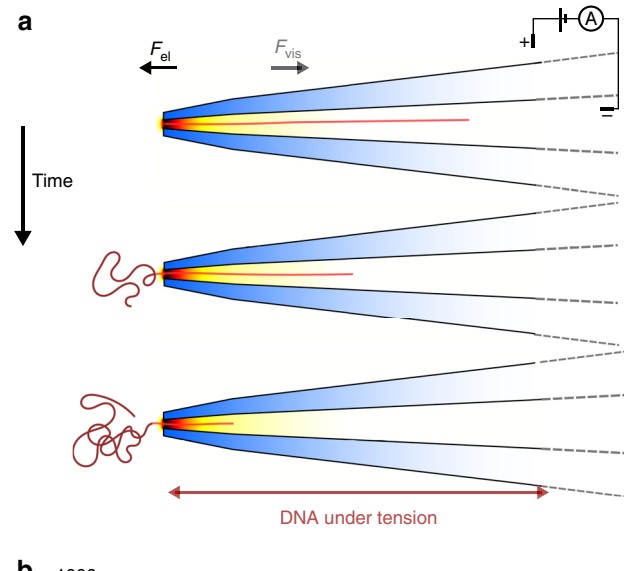

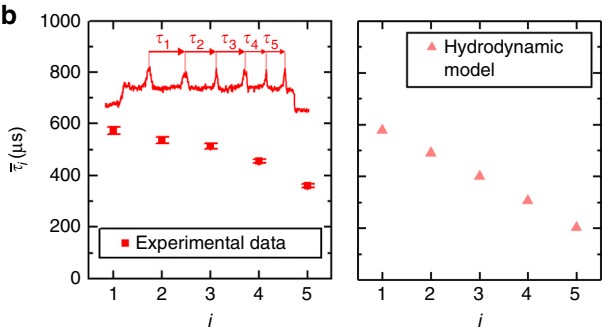

**Fig. 6 a** Schematic of translocation time course with DNA exiting from confinement. The portion of the DNA that has exited the nanopore buckles under a compressive load whereas the section of the DNA inside the nanopore is under tension. **b** Comparison of model with data on trajectory of DNA exiting nanopore. *Errors bars* show the standard error of the mean

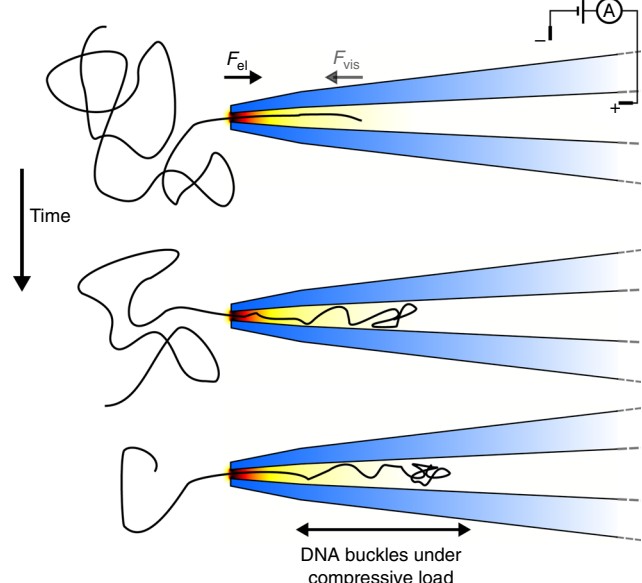

**Fig. 7** Schematic of translocation time course with DNA entering a conical nanopore. In this forward direction, the force vectors of the electric field and viscous drag are oppositely directed so the DNA buckles inside the conical nanopore and only a short section of DNA close to the entrance contributes to viscous drag

of the order the persistence length will contribute to viscous drag during the course of the translocation. Simple estimates based on Stokes drag on a cylinder show that viscous forces acting on this portion of DNA outside the pore are negligible compared to forces arising within the conical nanopore and so may be neglected[52]. Therefore, to determine the viscous drag, we need an estimate of $L$ for the part of the DNA within the nanopore. In both forward and backward translocations the net magnitudes of the electric and viscous forces within the nanopore are equal as required by force balance. However, the spatial distribution of the two forces are different. In a forward translocation the DNA is under a compressive load within the nanopore and in a backward translocation the DNA in the nanopore is under tension. These two situations are shown schematically in Figs. 6a and 7.

The relative importance of thermal noise compared to compressive applied loads may be characterised by considering a length scale that we shall call the buckling length, $l_B$. This is defined as the maximum length of the polymer that can remain straight under the applied load without undergoing the Euler buckling instability. The buckling length may be estimated as:

$$\frac{l_B}{l_P} = \left( \frac{8\beta kT}{l_P F_{tether}} \right)^{1/2}, \tag{8}$$

where $\beta$ is an order one parameter depending on the load distribution (Supplementary Note 2). Using a DNA persistence length of 30 nm (appropriate for DNA in 4 M LiCl[36]) and

$T = 23\,°C$ (the temperature used for experiments) gives $l_B/l_P \sim 0.8$, which shows that the DNA is liable to buckle under the compressive load soon after passing into the nanopore in the forward direction. This suggests that only a small constant length of DNA will likely contribute to viscous drag inside the conical part of the nanopore during translocation in the forward direction. In backward translocation the DNA is under tension and the effective length of the DNA can be approximated as the actual length residing within the conical nanopore at a given time.

We can therefore calculate the DNA trajectory in the backward direction by substituting $v = \frac{dL(t)}{dt}$ into Eq. (7), where $L(t)$ is the total length of DNA inside the cone at a certain timepoint of the translocation. The resulting differential equation can be integrated numerically (Supplementary Note 2). In Fig. 6b, we compare the predicted DNA trajectory with the experimental data. Importantly the model reproduces the observed trend of the data whereby the velocity continuously increases with time. The translocation timescale is in good agreement given the various approximations concerning the nanopore geometry. The velocity change is overestimated in the early phase of the translocation. This is most likely because the effective length of the DNA is in reality less than the contour length residing in the nanopore since not all the DNA will be pulled taut due to the widening aperture determined by the conical angle of the nanopore. This error becomes less significant as the translocation progresses and the contour length of DNA within the nanopore gets shorter.

An approximate analytical expression (Supplementary Note 2) can also be obtained for the dependence of the total translocation time in the backward direction $T_{back}$ with total DNA length $L_{max}$

$$T_{back} = \frac{\pi \mu a^2}{\alpha^2 F_{tether}} \frac{(\alpha L_{max}/a)^2}{\ln(\alpha L_{max}/2a)}. \tag{9}$$

This equation shows there is no algebraic scaling law however a local scaling can be defined as

$$n \equiv \frac{d(\ln T_{back})}{d(\ln L_{max})} = 2 - \frac{1}{\ln(\alpha L_{max}/2a)}. \tag{10}$$

If Eq. (10) is used for $L_{max} = 3–10$ kbp, we obtain $n = 1.69–1.77$ in fairly good accord with the measured slope of 1.5 (Fig. 3).

In the case of forward translocations the experimental data show a very slightly super-linear power law and the intra-event velocity has a small slow down of ~5%. This slow down indicates an increasing drag force which could be due to tension propagation along the part of the DNA outside the pore or an effect due to the crowding of DNA after it passes through the nanopore opening and subsequently buckles under the compressive load[2, 3, 53–56]. However the magnitude of the slow down is small and the velocity is constant to a good approximation. Within the constant velocity approximation, we can determine the effective length of DNA contributing to viscous drag by substituting a constant length $L = L_e$ in Eq. (7).

From the data on the forward direction in Fig. 4c, it takes on average 157 µs to cover 1032 bp i. e. $v = 2.2$ mm/s which yields $L_e = 385$ nm. This value represents the length of a straight rod with the same diameter as the DNA which would produce the observed velocity in the forward direction under an applied force of 3.4 pN.

## Discussion

In summary, we have used conically shaped nanopores to investigate the physics of how nanopore geometry can affect DNA translocation. A combination of length dependence studies, intra-event velocity determination using DNA rulers and optical tweezer measurements has allowed us to obtain a comprehensive experimental characterisation of the DNA dynamics. The intra-event velocity determination with DNA rulers represents a particularly powerful tool for examining the translocation process and could be used in the future for determining the dynamics for other nanopore geometries such as thin 2D membranes. We have shown here how a simple model can account for experimental results with asymmetric nanopores. The difference between the two cases is determined by the DNA configuration within the pore—in the backward case, the DNA is under tension and translocates as a straight rigid rod whereas in the forward case, it is under a compressive load and buckles resulting in a smaller, approximately time independent resistive force. Our improved understanding of DNA translocation dynamics will advance practical applications of solid-state nanopore based devices for reading information along the contour of a linear polyelectrolyte.

## Methods

**Nanopore fabrication and DNA ruler synthesis**. Conical glass nanopores were fabricated according to previously reported experimental protocols with a final diameter estimated at $14 \pm 3$ nm[16]. Experiments were performed in 4 M LiCl, 50 mM NaCl, 1 mM MgCl$_2$ electrolyte buffered at pH 8 with 10 mM Tris-HCl. Individual DNA lengths were chromatography-purified NoLimits DNA (ThermoFisher Scientific). The DNA ruler was synthesised by cutting circular 7249 base m13mp18 ssDNA (New England Biolabs) using the enzymes EcoRI and BamHI to form a linear ssDNA chain 7228 bases in length. The cut scaffold is then purified and mixed in a 1:5 ratio with 212 oligonucleotides which form a double-strand with six equidistant zones of dumbbell hairpins. The mixture is annealed in 10 mM MgCl$_2$ for 50 min before purification of the excess oligonucleotides using 100 kDa cut-off Amicon filters. All ionic current measurements were recorded using an Axopatch 200B amplifier with an external 8-pole Bessel filter at 50 kHz filtering frequency and then recorded at 250 kHz. For ping-pong experiments where an individual DNA molecule was threaded through the nanopore multiple times by reversing the voltage, a custom Labview 2013 (National Instruments) program was written whereby the voltage was switched 40 ms after the current deviated from a set threshold.

**Optical tweezers measurements**. The combined optical tweezers and nanopore set-up is the same as previously described by Otto et al.[57]. The homemade optical tweezers provide a stable 3D trap to capture streptavidin-coated polystyrene microparticles with ~2 µm diameter (Kisker, Germany). The colloids are coated with DNA rulers by using a biotin modification at the end of the DNA ruler. A high-speed camera (CMOS, MC1362, Mikrotron, Germany) captures the position of the bead and its displacement from the centre of the trap is converted into forces after calibrating the trap stiffness (power spectral density calibration method[58]). The relative pore-bead centre distance is controlled and varied with a piezoelectric nanopositioning system (P-517.3 and E-710.3, Physik Instrumente, Germany). The nanopore-ending quartz capillary is positioned between two reservoirs obtained by sealing two PDMS chambers onto a glass slide. The reservoirs are filled with the same salt solution used for the translocation experiments and in the tip-side chamber the DNA-functionalised colloids are added. Ionic currents were recorded with an Axopatch 200B and measurements controlled with a custom Labview program (National Instruments). The data were analysed with an automated IgorPro (WaveMetrics) program.

**Data availability**. The data that support the findings of this study are available on request from the corresponding author.

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

## Acknowledgements

The authors thank N. Laohakunakorn for useful discussions and initial optical tweezers work. N.A.W.B. and U.F.K. acknowledge funding from an ERC starting grant (Passmembrane 261101) and an ERC consolidator grant (Designerpores 647144). M.R. acknowledges funding from the Swiss National Science Foundation (Early Postdoc Mobility Fellowship). K.C. acknowledges funding from the Chinese Scholarship Council (201506210147).

## Author contributions

N.A.W.B., K.C., M.R. and U.F.K. planned the experiments, N.A.W.B., K.C. and M.R. performed the measurements; N.A.W.B. and S.G. developed the theoretical model and all authors wrote and edited the manuscript.

## Additional information

**Competing interests:** The authors declare no competing financial interests.

