## [Peer Review File · Nature Communications]

Reviewers' comments:

Reviewer #1 (Remarks to the Author):

Referee Report – Bell et al., “Asymmetric dynamics of DNA entering and exiting a strongly confining nanopore” - MS NCOMMS-16-30539

I have read the paper by Bell et al. that you asked me to comment on for Nature Communications. I enjoyed reading the paper. It reports on high quality experimental work and modeling for DNA translocating through a conically shaped nanopore. It is well written and I believe your journal should publish the manuscript. But I'd like the authors to address a few points either in the manuscript or in a separate reply to the editor.

First, I could not locate anywhere in the experimental part of the paper what the experimental values for the cone angles were for the nanopores studied. Indeed, to what approximation does the geometry of the pore actually look like a cone?

Second, Figure 3 shows translocation time data for 4 different pores with significant variation from pore to pore for backward translocation times. To what do the authors ascribe this effect? Is it variation in cone geometry?

Third, a value for cone angle is presented in the theoretical modeling. Where did this come from?

Fourth, it would be helpful to know how sensitive the model predictions are to cone angle so the reader can ascertain the geometrical range for which the asymmetry in pore geometry is important and how much the effects reported on could be increased further with smaller cone angles.

Finally, in several places in the paper the authors imply that slowing down DNA translocation speeds is important for DNA sequencing with solid-state nanopores. But never do the authors say by how much the DNA needs to be slowed down to make sequencing possible. I also fear that slowing down by increasing the drag alone may actually increase the fluctuations via the fluctuation dissipation theorem and therefore not improve the spatial resolution needed for sequencing. If the authors want to use sequencing as a justification for their work, they need to face the music on this issue.

Reviewer #2 (Remarks to the Author):

Summary

The authors show that double-stranded DNA can be detected with a micropipette and that the transport rate depends on the direction the molecule moves.

Review

The methods and experimental results appear to be sound. The manuscript is interesting and represents new findings.

I'm not convinced the cartoon in Fig. 1b is reasonable.

The results in Fig. 4 are particularly interesting. Only the backward time constants depend on the index and the rate increases as more of the polymer exits the pipette.

The theoretical model presented by the authors is reasonable. However, I'm not so sure the statement “In a forward translocation the DNA is under compression” is ok, in the engineering sense of the word (i.e., tension and compression of beams). Not in tension does not equate to

being under compression.

The authors state "Our improved understanding of DNA translocation dynamics will advance practical applications of solid-state nanopore based devices for reading information along the contour of a linear polyelectrolyte." I hope that is eventually borne out by experiment, but despite the long heralded promise of solid-state nanopores, they are still lacking with respect to DNA and polymer characterization determined with protein-based nanopores (see below).

The manuscript's scholarship is a bit lacking. The authors should put their work in context of the study of ion fluxes in micropipettes by Mathias, et al. (1990. Biophys J).

Citations to the earliest papers in the field of nanopore-based sensing (Bezrukov & Kasianowicz. 1993. Phys Rev Lett; Bezrukov, et al. 1996 Macromolecules), DNA sequencing (Kasianowicz, et al. 1996 PNAS; Manrao, et al. 2012. Nature Biotech; Kumar, et al. 2012. Sci. Reports; Fuller, et al. 2016. PNAS), and the separation of molecules based on their size (Robertson, et al. 2007. PNAS; Reiner, et al. 2010. PNAS; Baaken, et al., 2015. ACS Nano) should be included. Also, Henrickson, et al. (2010. J Chem Phys) showed that the details of voltage-dependent single-stranded DNA transport through a protein nanopore depended on the direction the molecules traveled through the pore. The geometry of that pore's vestibule presumably had some effect.

Reviewer #3 (Remarks to the Author):

In the manuscript, "Asymmetric dynamics of DNA entering and exiting a strongly confining nanopore", the authors present experimental results for the translocation of dsDNA into and out of a device they term a "conical nanopore". Experiments reveal that the dynamics are quite different between translocation into the device (called the forward direction) and out of the device (the backward direction). The timescales of translocation and the scaling are observed to be markedly different. To gain further insight, experiments are performed with a "DNA ruler" that has 6 hairpins equally distributed along the length of the DNA. These locations can be observed by distinct changes in the ion flux through the pore thus giving insight into the internal dynamics of translocation (as opposed to just start to finish). Results from these experiments reveal that the speed of translocation is constant in the forward direction, but significantly speeds up in the backward direction. Further insight is gained by attaching biotin to one end of the DNA and performing force measurements via optical tweezers. Finally, a relatively in-depth modeling of the system is developed and compared against the experimental results.

Overall the manuscript is very impressive. It is very satisfying and of significant usefulness to have standard translocation metrics (translocation time vs. DNA length), relatively new insight from the DNA ruler approach for the in-process dynamics, and force measurements from the optical tweezers experiments all for the same setup and conditions. The modeling is also noteworthy and a valuable contribution to the literature.

That being said, the manuscript does have some weaknesses that should be addressed. My main concern is that both the experimental and simulation results are quite surprising and the discussion regarding this is not very convincing. In fact, I find contradictions with another paper by the same authors on what appears to be the same system. While it is possible that I am missing something, these contradictions between relatively equivalent systems are very concerning to me. Further, the system setup is very interesting and useful, but it does not really conform to standard translocation setups. I find some of the statements to be too strong (We show that a hydrodynamic model, without adjustable fitting parameters, can account for these observations) and lacking citing previous work (Our results suggest that shaping the nanopore geometry offers a pathway for controlling the DNA translocation time). Further, the results are surprising – and it is questionable just how relevant they are to more standard setups.

I thus do not recommend publishing the manuscript in its current form and believe clarification on this points is necessary before a final decision is made.

A detailed discussion of these points is below.

Major points:

1. The term "conical nanopore"

In my experience with the translocation literature, a nanopore refers to a hole in a membrane with a thickness that is very small compared to the length of the polymer being studied and a conical nanopore is simply one that has a conical shape – that is, it is wedge-like when viewed in cross-section. For the system under study in this paper, the conical nanopore is quite long compared to the size of the dsDNA strand being studied. Thus, this can more accurately be described as translocation into or out of a long conical nanopore or perhaps translocation through a nanopore into/out of a tapered, cylindrical nanochannel. Indeed, these factors are considered in the modeling, but the terminology could easily be interpreted to indicate that this is a standard nanopore setup.

2. Applicability of the system

Following from point 1, the difference between this system and a standard nanopore setup are not trivial. The differences between the available space outside of the capillary and standard setups that correspond to a half-space are likely of little import, as indicated in the paper. However, the situation inside the device is significantly different than the usual case again consisting of a half-space. In comparison, DNA inside the capillary will experience significant confinement. Further, the taper angle of the capillary is small (0.05 radians) and thus the electric field in the axial directions decays much, much more slowly than in the standard setup. From the equations given in the paper, I calculate that for portions of the DNA 200 nm inside of the device, the axial field is still about 50% of what it is at the narrowest portion of the channel. At $x=450$ nm, it is 10%. This decay is much slower than in standard setups where the radial field decays as an exponential with a characteristic length proportional to the square of the diameter of the pore (see the work of Rowghanian and Grosberg).

This comparatively strong field and high degree of confinement are likely to affect the dynamics of translocation in either direction. In the forward direction, this field will continue to pull DNA away from the pore for significant distances. In a sense, this helps ameliorate a complication arising from the alternative: in the absence of this extended field, the high degree of confinement would yield significant crowding of the DNA post translocation that would very likely affect the dynamics. Nevertheless, I found discussion of these effects to be lacking in the paper. They are considered when modeling, but in my opinion, not enough to justify a conclusion that the results are directly relevant to standard translocation setups.

These considerations play an even bigger role in the backward direction. Here, considering the confinement and extended field, the DNA will start from a stretched conformation – which is quite different from typical models of translocation (As a counter-example that the authors could consider comparing against, see the work of G.W. Slater and D. Sean for translocation starting from a tube; further work has examined the effects of elongation of the polymer due to field gradients such as Farahpour and Ejtehadi as well as Vollmer and de Haan). The dynamics in this direction are of great interest – the authors see the translocation time is lower than in the other direction – as predicted/observed by others. However, the insight that these results give for the classic nanopore-translocation setup is not clear.

2. Surprising experimental results

Concerning the experimental results for forward translocation, I find the results very surprising. Specifically, the authors find a constant translocation speed. As discussed in the paper, this contradicts the predictions of the tension-propagation model where the speed would be expected to decrease as the tension front travels down the polymer and then decrease once it reaches the end. While it may not be possible to resolve these details, finding a constant velocity is very surprising to me. The authors discuss this in the context of finite size effects and a length independent pore friction that overwhelms the drag friction that would grow as the tension travels down the polymer. First, the authors give the Zimm relaxation time to be 8 microseconds and so with translocation times on the order of 2 ms in the forward direction, this process is decidedly not

quasi-static and thus tension propagation would be expected to play a role. Second, the I am having trouble coming up with a source of pore friction that could overwhelm the drag force – the polymers are not that short. In the cited theoretical work by Ikonen et al., the pore friction is a very local effect that is only dominant for a couple of monomers before drag takes over. The authors claim that the increasing drag force would be a factor only for “extremely long” polymer lengths – I don’t believe this to be the case and would expect to see such effects at the current lengths. Thus I find the discussion of the rather surprising results unconvincing.

Further, on examining a paper by the authors on arXiv that is cited in the current manuscript, results for a very similar length of DNA (7 kbp vs. 8 kbp) do show a decrease in the translocation speed as translocation progresses. In fact this is done for several pores and this result is consistently found. Note that the pores have the same pore diameter as the one given in this study, the applied voltage is of a similar value, the salt type and concentration are the same, and the net translocation times are also the same.

In fact, the paper on arXiv discusses this result in the context of an increasing drag force (as follows from tension propagation), discusses the processes being non-equilibrium due to the relaxation time being much longer than the translocation time. To be clear, it seems that tension propagation is an acknowledged effect at 7kbp whereas in the current manuscript considering 8 kbp strands it was relegated to “extremely long” polymer lengths. Further, the arXiv paper also discusses the possible effects of the confinement in the capillary for translocation in the forward direction and discusses how this effect is often neglected as it is assumed that it buckles under compression – is this buckling under compression not the model used in the manuscript under review?

Although it is possible there is an underlying difference that I am not seeing, right now this seems like a contradiction to me in terms of both the experimental data and the interpretation and thus some clear explanation is necessary. While the difference in the data can perhaps be explained by different scales on the plots (I am guessing), the interpretations and physical pictures are not consistent and this is confusing at best.

3. Scaling

The scaling of the translocation time with polymer length – a standard measurement for translocation work – is very different in the forward and backward directions going from 1.1 to 1.5. A physical picture explaining this marked difference would be very welcome. Again, some previous theoretical/simulation work has examined the effect of starting translocation from a tube and how this effects the scaling of the translocation time. Further, another major result from that work is how the relative variance in the translocation times (ie, the variance normalized by the translocation times) can be reduced depending on the geometry of the tube

4. Insight from Model

Following from point 3, the model would seem to be a good source of corroboration of the physical picture. However, the predictions of the model are somewhat confusing. In the forward direction, a scaling exponent of 1.0 is obtained. While this is not far off the exponent measured from the experiments, it is a bit concerning that an exponent of precisely 1.0 is obtained. This would correspond to limit where translocation is fully driven (ie, no thermal fluctuations) and friction is entirely a local effect at the pore and thus length independent. This does not seem physical.

In the backward direction, the exponent rise quite rapidly to values significantly greater than 1.5 and yields values in the range of 1.7 to 1.8 for the DNA lengths studied. In fact, in the long polymer limit, an exponent of 2.0 is predicted. This is in a sense the opposite limit of the exponent predicted in the forward direction as it corresponds to the purely diffusive limit where the diffusion coefficient is determined locally (ie, length independent) as predicted in the early theoretical work of both Muthukumar and Sung and Park.

Can the authors give insight into these somewhat surprising predictions?

Further, in the forward direction, the observed translocation velocity is used to determine a characteristic length of 227 nm. What is the relevance of this number? It is just stated and it is not

clear what value it has.

Overall, the modeling approach is useful, but the results of the model are questionable for a number of reasons: i) the applicability to standard nanopore setups is unclear, ii) the scaling predictions of the model seem unphysical, and iii) the meaning of the characteristic length in the forward direction is not discussed.

Smaller points:

1. Would Fig. 3 be more useful as a log-log plot?
2. For the force measurement shown in figure 5 b at $V=400$ mV, there seems to be a noticeable decrease in the force during the pulling at a constant speed. Can this be related to the decreasing length of the polymer inside of the nanocapillary which corresponds to a decrease in the net force opposing the retraction?
3. The authors state that "Using the ping-pong format (rather than a constant potential) enabled us to determine the DNA length from the ECD in the forward direction" – is this because of the enhanced statistics from multiple events or is there another reason?
4. The analysis is limited to events that display no hairpins during translocation. However, as seen in the supplemental material, these are only $\sim 30\%$ of the events in the forward direction and $\sim 10\%$ of the events in the backward direction. Hence, there seems to be a distinct possibility of a selection bias in the data. Specifically, only events that thread into the pore by an end are studied – and these are likely to represent a population of events in which the polymer is somewhat elongated prior to translocation (in the sense that the DNA near the pore is stretched out – again see Farahpour and Ejtehadi as well as Vollmer and de Haan). Have the authors considered the possible effects of such a selection bias.

We thank the reviewers for their comments and suggestions which we believe have helped to strengthen the paper. Below we have answered each comment in turn.

We have found that our force spectroscopy data used an incorrect calibration value for the optical trap stiffness so that the forces shown previously were 0.65x the values they should have been. We have now updated all graphs in the paper using the force data (Figures 5, 6b) and the values given in Tables. This does not change the findings or interpretation of the data in the text.

Reviewers' comments:

Reviewer #1 (Remarks to the Author):

I have read the paper by Bell et al. that you asked me to comment on for Nature Communications. I enjoyed reading the paper. It reports on high quality experimental work and modeling for DNA translocating through a conically shaped nanopore. It is well written and I believe your journal should publish the manuscript. But I'd like the authors to address a few points either in the manuscript or in a separate reply to the editor.

We appreciate the reviewer's comments on the quality of the work and have provided a detailed response to specific points below.

First, I could not locate anywhere in the experimental part of the paper what the experimental values for the cone angles were for the nanopores studied. Indeed, to what approximation does the geometry of the pore actually look like a cone?

We have now included a typical scanning electron microscope image in Figure 1 to show that the pore geometry is indeed very well described by a cone.

Updated Figure 1

The values given for the pore diameter and cone semi-angle are average values from scanning electron microscopy images of a sample of 20 pores made using our fabrication method. This data was collected in a previous paper; Bell, N. A. W. *et al Nat. Nanotechnol.* doi:10.1038/nano.2016.50, 2016 which used the same fabrication process. We have updated a sentence where we give the nanopore diameter to also include the semi-cone angle; "The diameter of the nanopores used was

estimated as 14 ± 3 nm (mean \pm s.d.) with a cone semi-angle angle of 0.05 ± 0.01 radians (mean \pm s.d.) based on previous characterisation of the fabrication process [Bell *et al.*, NNano, 2016].”

Second, Figure 3 shows translocation time data for 4 different pores with significant variation from pore to pore for backward translocation times. To what do the authors ascribe this effect? Is it variation in cone geometry?

Yes, we ascribe this effect to small variations in pore geometry. All other parameters – salt concentration, voltage, DNA lengths do not change and therefore differences in pore geometry must account for the different translocation times. It is well known from other work on solid-state nanopores (see eg Plesa *et al*, Velocity of DNA during translocation through a solid-state nanopore, 2014, Nano Lett) that the translocation velocity can vary according to small differences in 3D pore geometry.

We have added a sentence to say that the small variation is likely due to nanopore geometry differences:

“The translocation time, for a fixed direction and length, varies slightly from one pore to the next which we attribute to small differences in the pore geometry.”

Third, a value for cone angle is presented in the theoretical modeling. Where did this come from?

See the reply to the first comment.

Fourth, it would be helpful to know how sensitive the model predictions are to cone angle so the reader can ascertain the geometrical range for which the asymmetry in pore geometry is important and how much the effects reported on could be increased further with smaller cone angles.

In the SI we have now included a graph which shows how the model predictions for the intra-event times vary if we change the semi-cone angle by \pm one standard deviation from the mean value and we have also plotted this variation for the pore radius.

In general smaller values of semi-cone angle and pore radius decrease the velocity since the closer proximity of the nanopore walls to the DNA increases viscous drag.

Finally, in several places in the paper the authors imply that slowing down DNA translocation speeds is important for DNA sequencing with solid-state nanopores. But never do the authors say by how much the DNA needs to be slowed down to make sequencing possible. I also fear that slowing down by increasing the drag alone may actually increase the fluctuations via the fluctuation dissipation theorem and therefore not improve the spatial resolution needed for sequencing. If the authors want to use sequencing as a justification for their work, they need to face the music on this issue.

We agree with the reviewer that slowing down the DNA does not necessarily imply an increase in resolution since fluctuations must also be taken into account. Indeed, as we show in Figure 2, although the backward DNA translocation is slower, the scatter of data points is significantly larger compared to the forward direction showing that there are much greater fluctuations in the backward direction. Slowing down the translocation should still improve signal to noise however since there will

be less filter attenuation of the signal. To avoid any confusion that slowing down necessarily implies greater resolution, we have removed the following sentence from the abstract:

“Our results suggest that shaping the nanopore geometry offers a pathway for controlling the DNA translocation time.”

And also in the conclusion we have removed the sentence:

“Furthermore our results demonstrate that tuning the configurational constraints on translocating DNA provides an avenue for controlling the DNA velocity.”

And in the introduction

“There is also a deficit in our understanding of how different nanopore geometries can affect translocation dynamics and how the geometry can be best tuned to reduce translocation speed thereby potentially improving resolution.”

Was replaced with

“There is also a deficit in our understanding of how different nanopore geometries can affect translocation dynamics and which geometry gives the highest resolution possible for reading information along the DNA contour.”

As we state in the introduction, the main justification of the work was to provide a comprehensive experimental and modelling study of DNA transport in a strongly confining asymmetric nanopore geometry.

Reviewer #2 (Remarks to the Author):

Summary

The authors show that double-stranded DNA can be detected with a micropipette and that the transport rate depends on the direction the molecule moves.

Review

The methods and experimental results appear to be sound. The manuscript is interesting and represents new findings.

We appreciate that the reviewer finds the results novel and interesting.

I'm not convinced the cartoon in Fig. 1b is reasonable. The results in Fig. 4 are particularly interesting. Only the backward time constants depend on the index and the rate increases as more of the polymer exits the pipette.

The theoretical model presented by the authors is reasonable. However, I'm not so sure the statement “In a forward translocation the DNA is under compression” is ok, in the engineering sense of the word (i.e., tension and compression of beams). Not in tension does not equate to being under compression.

We agree that the phrase “DNA is under compression” was not an accurate description. The DNA is subject to a compressive load and undergoes a buckling transition precisely to avoid compression which is not energetically favourable. We have changed “under compression” to “under a compressive load” in all instances where we used this phrase in the manuscript.

The manuscript's scholarship is a bit lacking. The authors should put their work in context of the study of ion fluxes in micropipettes by Mathias, et al. (1990. Biophys J).

Mathias *et al.* studied the flux of ions between a micropipette and a cell in a whole cell patch clamp experiment. We have now included a reference to this work and others which previously studied transport through conical nanopores:

“We used conically shaped nanopores fabricated by laser-assisted capillary pulling. Such conical pores have previously been used to study the transport of ions {Mathias1990, Chen2017} and macromolecules {Li2013,Thacker2012,Laohakunakorn2013} in confinement.”

Citations to the earliest papers in the field of nanopore-based sensing (Bezrukov & Kasianowicz. 1993. Phys Rev Lett; Bezrukov, et al. 1996 Macromolecules), DNA sequencing (Kasianowicz, et al. 1996 PNAS; Manrao, et al. 2012. Nature Biotech; Kumar, et al. 2012. Sci. Reports; Fuller, et al. 2016. PNAS), and the separation of molecules based on their size (Robertson, et al. 2007. PNAS; Reiner, et al. 2010. PNAS; Baaken, et al., 2015. ACS Nano) should be included. Also, Henrickson, et al. (2010. J Chem Phys) showed that the details of voltage-dependent single-stranded DNA transport through a protein nanopore depended on the direction the molecules traveled through the pore. The geometry of that pore’s vestibule presumably had some effect.

We have added all these citations at the relevant points in the introduction. We have also added a comment on the direction dependent behaviour of ssDNA transport through alpha-hemolysin described by Henrickson *et al.*: “The geometry of the pore is known to also play a role in the transport dynamics of single stranded DNA through biological pores. For instance, the DNA capture rate and current signature during passage through α -hemolysin depends on the direction of transport due to the pore’s structural asymmetry {Henrickson2000,Henrickson2010}.”

Reviewer #3 (Remarks to the Author):

In the manuscript, “Asymmetric dynamics of DNA entering and exiting a strongly confining nanopore”, the authors present experimental results for the translocation of dsDNA into and out of a device they term a “conical nanopore”.

Overall the manuscript is very impressive. It is very satisfying and of significant usefulness to have standard translocation metrics (translocation time vs. DNA length), relatively new insight from the DNA ruler approach for the in-process dynamics, and force measurements from the optical tweezers experiments all for the same setup and conditions. The modeling is also noteworthy and a valuable contribution to the literature.

We would like to thank the referee for this encouraging and positive assessment of the manuscript. We also appreciate the many insightful suggestions that we address in our detailed response below. As described already in our response to reviewer 1 we have removed the phrase “Our results suggest that shaping the nanopore geometry offers a pathway for controlling the DNA translocation time” and also have also removed the phrase “without adjustable fitting parameters” in response to the reviewer’s suggestion.

Major points:

1. The term “conical nanopore”

In my experience with the translocation literature, a nanopore refers to a hole in a membrane with a thickness that is very small compared to the length of the polymer being studied and a conical nanopore is simply one that has a conical shape – that is, it is wedge-like when viewed in cross-section. For the system under study in this paper, the conical nanopore is quite long compared to the size of the dsDNA strand being studied. Thus, this can more accurately be described as translocation into or out of a long conical nanopore or perhaps translocation through a nanopore into/out of a tapered, cylindrical nanochannel. Indeed, these factors are considered in the modeling, but the terminology could easily be interpreted to indicate that this is a standard nanopore setup.

In the abstract we have changed the term “conical confinement” to “extended conical confinement” to try to indicate that the pore is longer compared to nanopores in thin membranes for those who read only the abstract. We now also have the electron microscopy image in Figure 1 (see response to reviewer 1) which now clearly conveys the geometry of this system.

2. Applicability of the system

Following from point 1, the difference between this system and a standard nanopore setup are not trivial. The differences between the available space outside of the capillary and standard setups that correspond to a half-space are likely of little import, as indicated in the paper. However, the situation

inside the device is significantly different than the usual case again consisting of a half-space. In comparison, DNA inside the capillary will experience significant confinement. Further, the taper angle of the capillary is small (0.05 radians) and thus the electric field in the axial directions decays much, much more slowly than in the standard setup. From the equations given in the paper, I calculate that for portions of the DNA 200 nm inside of the device, the axial field is still about 50% of what it is at the narrowest portion of the channel. At $x=450$ nm, it is 10%. This decay is much slower than in standard setups where the radial field decays as an exponential with a characteristic length proportional to the square of the diameter of the pore (see the work of Rowghanian and Grosberg).

This comparatively strong field and high degree of confinement are likely to affect the dynamics of translocation in either direction. In the forward direction, this field will continue to pull DNA away from the pore for significant distances. In a sense, this helps ameliorate a complication arising from the alternative: in the absence of this extended field, the high degree of confinement would yield significant crowding of the DNA post translocation that would very likely affect the dynamics. Nevertheless, I found discussion of these effects to be lacking in the paper. They are considered when modeling, but in my opinion, not enough to justify a conclusion that the results are directly relevant to standard translocation setups.

These considerations play an even bigger role in the backward direction. Here, considering the confinement and extended field, the DNA will start from a stretched conformation – which is quite different from typical models of translocation (As a counter-example that the authors could consider comparing against, see the work of G.W. Slater and D. Sean for translocation starting from a tube; further work has examined the effects of elongation of the polymer due to field gradients such as Farahpour and Ejtehadi as well as Vollmer and de Haan). The dynamics in this direction are of great interest – the authors see the translocation time is lower than in the other direction - as predicted/observed by others. However, the insight that these results give for the classic nanopore-translocation setup is not clear.

We agree with the reviewer that there are important differences between the setup used here and a nanopore in a thin 2D membrane. Indeed it is not our intention to argue that the results here would apply to thin 2D membranes. Firstly there should be no direction dependence for translocations across a 2D membrane since the pore has two symmetric half spaces. This has in fact been measured by Gershow *et al.* (NNano, 2007, their SI Fig. 1) and we have now added a sentence to describe this.

“In contrast to the results shown here for conical nanopores, ping-pong experiments with nanopores in thin 2D membranes show the same translocation time irrespective of which direction the DNA is driven through as expected for two symmetric semi-infinite spaces {Gershow2007}.”

We also agree with the reviewer that of the two translocation directions, the backward direction is clearly the most different to the “classic” thin membrane translocation problem since the starting DNA configurations are strongly limited by the confinement of the slowly tapering cone. The forward direction is more similar to the classic problem since the starting configurations are of a random coil. As the reviewer says there are still potential differences due to the longer range electric field and crowding due to the geometric confinement compared to the thin membrane case. Although it should be noted that the electric field is still short (50% over 200nm) compared to the DNA length eg 7kbp=2400nm. Overall we believe that the way forward to address this issue is by performing experiments that compare the in-process dynamics of conical vs thin membrane geometries (there is no published experimental data on the in-process dynamics for thin membrane pores). Since this is beyond the scope of the current work we have also added a sentence in the summary to point to future work which should provide this comparison:

“The intra-event velocity determination with DNA rulers represents a particularly powerful tool for examining the translocation process and could be used in the future for determining the dynamics for other nanopore geometries such as thin 2D membranes.”

2. Surprising experimental results

Concerning the experimental results for forward translocation, I find the results very surprising. Specifically, the authors find a constant translocation speed. As discussed in the paper, this contradicts the predictions of the tension-propagation model where the speed would be expected to decrease as the tension front travels down the polymer and then decrease once it reaches the end.

While it may not be possible to resolve these details, finding a constant velocity is very surprising to me. The authors discuss this in the context of finite size effects and a length independent pore friction that overwhelms the drag friction that would grow as the tension travels down the polymer. First, the authors give the Zimm relaxation time to be 8 microseconds and so with translocation times on the order of 2 ms in the forward direction, this process is decidedly not quasi-static and thus tension propagation would be expected to play a role. Second, the I am having trouble coming up with a source of pore friction that could overwhelm the drag force – the polymers are not that short. In the cited theoretical work by Ikonen et al., the pore friction is a very local effect that is only dominant for a couple of monomers before drag takes over. The authors claim that the increasing drag force would be a factor only for “extremely long” polymer lengths – I don’t believe this to be the case and would expect to see such effects at the current lengths. Thus I find the discussion of the rather surprising results unconvincing.

Further, on examining a paper by the authors on arXiv that is cited in the current manuscript, results for a very similar length of DNA (7 kbp vs. 8 kbp) do show a decrease in the translocation speed as translocation progresses. In fact this is done for several pores and this result is consistently found. Note that the pores have the same pore diameter as the one given in this study, the applied voltage is of a similar value, the salt type and concentration are the same, and the net translocation times are also the same.

In fact, the paper on arXiv discusses this result in the context of an increasing drag force (as follows from tension propagation), discusses the processes being non-equilibrium due to the relaxation time being much longer than the translocation time. To be clear, it seems that tension propagation is an acknowledged effect at 7kbp whereas in the current manuscript considering 8 kbp strands it was relegated to “extremely long” polymer lengths. Further, the arXiv paper also discusses the possible effects of the confinement in the capillary for translocation in the forward direction and discusses how this effect is often neglected as it is assumed that it buckles under compression – is this buckling under compression not the model used in the manuscript under review?

Although it is possible there is an underlying difference that I am not seeing, right now this seems like a contradiction to me in terms of both the experimental data and the interpretation and thus some clear explanation is necessary. While the difference in the data can perhaps be explained by different scales on the plots (I am guessing), the interpretations and physical pictures are not consistent and this is confusing at best.

We thank the reviewer for these points and the opportunity to clarify the situation. The pre-print we have on ArXiv describes a separate study which is mainly concerned with the nature of fluctuations of DNA as it translocates in the forward direction through a conical nanopore. The data is consistent - we have made the Figure below for the reviewer to show that when plotted on the same axis range as the ArXiv paper, the experimental data in both studies show a small slow down.

To show this more clearly in Figure 4c of the current paper we have now added an inset which shows the forward translocation data with an appropriate resolution so that the error bars are visible. The error bars are much smaller in the forward direction compared to the backward direction since the fluctuations are less (see Figure 2a).

Updated Figure 4c with inset showing forward direction data at greater resolution.

Therefore we do consistently observe a small slow down in the forward direction of ~5% and there is no contradiction in the experimental data. We do currently say this in the Kinematics of Translocation experimental section of the paper in the sentence “It is also evident from Figure 4b that in the backward direction the DNA tends to speed up significantly whereas in the forward direction the velocity remains approximately constant albeit with a small increase of ~5% from τ_1 to τ_5 in agreement with previous results (citation to ArXiv paper).” but clearly this was not presented adequately. We hope that the update prevents any confusion.

Although we observe this small ~5% decrease in the forward direction our model for the translocation process is of a constant friction as the DNA moves through. The reason we did this is that we believe it is reasonable to model a constant velocity process given the small magnitude of the change in the forward direction given that there is a much noticeably greater velocity change in the backward direction. We agree with the reviewer that we did not make this explicitly clear in the current version of the paper and have now rewritten the model discussion as follows:

“In the case of forward translocations the experimental data show a very slightly super-linear power law and the intra-event velocity has a small slow down of ~5%. This slow down indicates an increasing drag force which could be due to tension propagation along the part of the DNA outside the pore or an effect due to the crowding of DNA after it passes through the nanopore opening and subsequently buckles under the compressive load {Grosberg2006,Sakaue2010,Lu2011,Palyulin2014,Panja2013,Ikonen2012}. However the magnitude of the slow down is small and the velocity is constant to a good approximation.”

3. Scaling

The scaling of the translocation time with polymer length – a standard measurement for translocation work – is very different in the forward and backward directions going from 1.1 to 1.5. A physical picture explaining this marked difference would be very welcome. Again, some previous theoretical/simulation work has examined the effect of starting translocation from a tube and how this affects the scaling of the translocation time. Further, another major result from that work is how the relative variance in the translocation times (ie, the variance normalized by the translocation times) can be reduced depending on the geometry of the tube

We have made a combined answer to points 3 and 4 – see below.

4. Insight from Model

Following from point 3, the model would seem to be a good source of corroboration of the physical picture. However, the predictions of the model are somewhat confusing. In the forward direction, a scaling exponent of 1.0 is obtained. While this is not far off the exponent measured from the experiments, it is a bit concerning that an exponent of precisely 1.0 is obtained. This would correspond to limit where translocation is fully driven (ie, no thermal fluctuations) and friction is entirely a local effect at the pore and thus length independent. This does not seem physical. In the backward direction, the exponent rise quite rapidly to values significantly greater than 1.5 and yields values in the range of 1.7 to 1.8 for the DNA lengths studied. In fact, in the long polymer limit, an exponent of 2.0 is predicted. This is in a sense the opposite limit of the exponent predicted in the forward direction as it corresponds to the purely diffusive limit where the diffusion coefficient is

determined locally (ie, length independent) as predicted in the early theoretical work of both Muthukumar and Sung and Park.

Can the authors give insight into these somewhat surprising predictions?

The works of both Muthukumar and Sung and Park consider the quasi-equilibrium theory of polymer translocation where the polymer chain is equilibrated at every step of translocation (ie translocation time \gg Zimm time). As we describe in the paper (the reviewer also refers to this earlier) in our experiments the translocation is decidedly non-equilibrium since the Zimm time is longer than the translocation time and hence quasi-equilibrium theories of Muthukumar, Sung and Park are not expected to be applicable.

Our theory is based on a hydrodynamic model to calculate the DNA velocity from the balance of electric and viscous drag forces. It has previously been shown that this approach can accurately model the speed of DNA passing through a thin 2D membrane nanopore (see Ghosal, S. Effect of Salt Concentration on the Electrophoretic Speed of a Polyelectrolyte through a Nanopore. *Phys. Rev. Lett.* **98**, 238104 (2007)). Our model then explains the difference in the forward and direction scaling for asymmetric pores as follows:

- In both the forward and backward direction the electric force on the DNA is determined by a short length (relative to the contour length of the DNA) close to the nanopore tip.
- In the backward direction the DNA is under tension and must have an initial configuration which is straightened out due to the conical confinement. Therefore there is a time dependent drag which we can be approximated as being equal to the length of the DNA remaining inside the pore. Calculating the mathematics of this model does not result in a unique scaling law but a local one can be defined as described in the text. In the long polymer limit the scaling is 2.0 as expected for a linear polymer being adsorbed into a hole see Grosberg, A. Y., Nechaev, S., Tamm, M. & Vasilyev, O. How long does it take to pull an ideal polymer into a small hole? *Phys. Rev. Lett.* **96**, 2–5 (2006).
- In the forward direction we observe a small slow down (potentially due to tension propagation or crowding effects) but this is a minor effect – only ~5%. This observation points to a close to constant friction on the DNA inside the nanopore which can be rationalised from the fact that the DNA is under compression here so will have a tendency to buckle. The buckled portion of the DNA will not be straight and therefore contribute little to viscous drag so that only a small constant length of DNA close to the pore entrance will be the source of drag and the scaling will be close to 1.0.

Finally we have now included a Figure in the SI comparing the distribution of ECD in the forward and backward directions (ECD is known to be linearly proportional to translocation time). This shows that the relative width of the distribution is larger in the backward direction (exiting confinement). Our model does not address the distribution widths but we have included this in response to the reviewer's suggestion.

Figure S3 – Comparison of distribution of event charge deficit values for forward and backward directions. The data is the same as presented in Figure 2a in the main text. The values show the parameters of Gaussian functions fitted to the distributions.

Further, in the forward direction, the observed translocation velocity is used to determine a characteristic length of 227 nm. What is the relevance of this number? It is just stated and it is not clear what value it has.

This value simply represents the length of a rigid rod which would reproduce the observed velocity of the DNA (with the correction to the force value of 2.2 pN to 3.4 pN this value is now 382 nm). Essentially it is a phenomenological value based on the observation that the velocity only shows a ~5% change so is well approximated by a time independent drag. We have added a sentence here to explain this:

“This value represents the length of a straight rod with the same diameter as the DNA which would produce the observed velocity in the forward direction under an applied force of 3.4 pN.”

Overall, the modeling approach is useful, but the results of the model are questionable for a number of reasons: i) the applicability to standard nanopore setups is unclear, ii) the scaling predictions of the model seem unphysical, and iii) the meaning of the characteristic length in the forward direction is not discussed.

We hope that our comments above have answered these questions.

Smaller points:

1. Would Fig. 3 be more useful as a log-log plot?

We have now plotted Figure 3 on log-log scales as suggested by the reviewer:

Updated Figure 3

2. For the force measurement shown in figure 5 b at V=400 mV, there seems to be a noticeable decrease in the force during the pulling at a constant speed. Can this be related to the decreasing length of the polymer inside of the nanocapillary which corresponds to a decrease in the net force opposing the retraction?

Yes, the force decreases in the last section as the DNA is pulled out since the electric field is spread over several hundred nanometres. This was in fact studied previously by Bulushev, R. D. *et al. Nano Lett.* 14, 6606–6613 (2014). We have added a sentence in the Figure caption together with a reference to this paper.

“The force decreases slightly towards the end of the trace since the DNA no longer occupies the high electric field zone which extends a few hundred nanometres from the pore entrance \cite{Bulushev2014}.”

3. The authors state that “Using the ping-pong format (rather than a constant potential) enabled us to determine the DNA length from the ECD in the forward direction” – is this because of the enhanced statistics from multiple events or is there another reason?

In the backward direction there is significantly greater scatter in the translocation statistics (see Figure 2a). This means that the statistics of the four DNA lengths used (3, 5, 7, 10 kbp) strongly overlap so we can't tell them apart. However in the forward direction, due to the smaller scatter, we can easily distinguish the lengths (Bell, N. A. W., Muthukumar, M. & Keyser, U. F. *Phys. Rev. E* **93**, 22401 (2016). Demonstrated this previously). So if we use the ping-pong format we can tell what the DNA length is from the forward translocation. We have added the underlined section of this sentence:

“Using the ping-pong format (rather than a constant potential) enabled us to determine the DNA length from the event charge deficit in the forward direction \cite{Bell2016} - the event scatter in the backward direction being too broad to be able to identify the different DNA lengths.”

4. The analysis is limited to events that display no hairpins during translocation. However, as seen in the supplemental material, these are only ~ 30% of the events in the forward direction and ~10% of the events in the backward direction. Hence, there seems to be a distinct possibility of a selection bias in the data. Specifically, only events that thread into the pore by an end are studied – and these are likely to represent a population of events in which the polymer is somewhat elongated prior to translocation (in the sense that the DNA near the pore is stretched out – again see Farahpour and Ejtehadi as well as Vollmer and de Haan). Have the authors considered the possible effects of such a selection bias.

We do select for the unfolded DNA configurations (ie current traces where there are no spikes indicating folds or knots). However these are the appropriate configurations for our model which only considers these so-called “type 1” translocations. We agree with the referee that it may be interesting to study the physics of folded configurations in the future but we believe that the discussion is much clearer by concentrating on the case of unfolded events.

REVIEWERS' COMMENTS:

Reviewer #1 (Remarks to the Author):

I have reviewed the changes made to Manuscript #NCOMMS-16-30539A by Bell et al. I find that the changes made significantly enhance the manuscript. In my opinion it is now ready for publication.

Reviewer #2 (Remarks to the Author):

The authors have addressed the issues raised by me. However, I would like them to further clarify some of the text in the current version of the manuscript. Specifically, the authors state "There is also a deficit in our understanding of how different nanopore geometries can affect translocation dynamics and which geometry gives the highest resolution possible for reading information along the DNA contour". What do they mean by "reading information" and if it is the sequencing of intact double stranded DNA, how do they propose to do that?

Reviewer #3 (Remarks to the Author):

The authors have adequately addressed my concerns and I recommend publication. I appreciate the detailed response and believe there are many continued, fruitful discussions to be had, but I think the paper now appropriately indicates this.

I found one typo in the response:

"velocity is constant to a good approximation"

We thank the reviewers for their further comments which we have replied to below.

Reviewer #2 (Remarks to the Author):

The authors have addressed the issues raised by me. However, I would like them to further clarify some of the text in the current version of the manuscript. Specifically, the authors state “There is also a deficit in our understanding of how different nanopore geometries can affect translocation dynamics and which geometry gives the highest resolution possible for reading information along the DNA contour”. What do they mean by “reading information” and if it is the sequencing of intact double stranded DNA, how do they propose to do that?

By reading information we mean the positions of bound objects along double-stranded DNA. For instance PNA has been shown to report on specific internal sequences of dsDNA to enable barcode identification and also there is strong interest in determining the positions of dsDNA binding proteins on DNA samples. We added a citation to PNA work and a paper of ours that measures proteins binding to DNA to the sentence which mentions reading information:

“There is also a deficit in our understanding of how different nanopore geometries can affect translocation dynamics and which geometry gives the highest resolution possible for reading information along the DNA contour {Singer2012, Bell2016b}.”

Reviewer #3 (Remarks to the Author):

The authors have adequately addressed my concerns and I recommend publication. I appreciate the detailed response and believe there are many continued, fruitful discussions to be had, but I think the paper now appropriately indicates this.

I found one typo in the response:
"velocity is constant to a good approximation"

There is no typo here to the best of our knowledge.